# An Adaptive Routing Algorithm for Inter-Satellite Networks Based on the Combination of Multipath Transmission and Q-Learning

Yuanji Shi , Zhiwei Yuan, Xiaorong Zhu and Hongbo Zhu *

College of Telecommunications and Information Engineering, Nanjing University of Posts and Telecommunications, Nanjing 210003, China
* Correspondence: zhb@njupt.edu.cn

**Abstract:** In a satellite network, the inter-satellite link can facilitate the information transmission and exchange between satellites, and the packet routing of the inter-satellite link is the key development direction of satellite communication systems. Aiming at the complex topology and dynamic change in LEO satellite networks, the traditional single shortest path algorithm can no longer meet the optimal path requirement. Therefore, this paper proposes a multi-path routing algorithm based on an improved breadth-first search. First, according to the inter-satellite network topology information, the improved breadth-first search algorithm is used to obtain all the front hop node information of the destination node. Second, all the shortest paths are obtained by backtracking the path through the front hop node. Finally, according to the inter-satellite network, the bandwidth capacity of the traffic and nodes determines the optimal path from multiple shortest paths. However, due to the high dynamics of low-orbit satellite networks, the topology changes rapidly, and the global topology of the network is often not available. At this time, in order to enhance the adaptability of the algorithm, this paper proposes an inter-satellite network dynamic routing algorithm based on reinforcement learning. Verified by simulation experiments, the proposed algorithm can improve the throughput of the inter-satellite network, reduce the time delay, and the packet loss rate.

**Keywords:** inter-satellite network; multi-path routing; Q-learning



## 1. Introduction

In recent years, countries around the world have been planning and building a Low-Earth-Orbit (LEO) satellite mobile communication system. The satellites in LEO have been moving at a high-speed relative to the ground, and seamless coverage requires a large number of satellites to be deployed in LEO, so the interconnection between satellites has become one of the key requirements of LEO satellite communication systems. On the other hand, since the control and measurement of the ground gateway station is required for the satellite communication system operation, and a large number of satellites are always moving at high speed, the interconnection between satellites is particularly important in order to ensure that the high-speed moving satellites can always maintain a link with the gateway station in the fixed area on the ground. As one of the key development directions in satellite communication systems, inter-satellite link technology can interconnect multiple satellites through inter-satellite links to realize information transmission and exchange between satellites and form a space communication with satellites as exchange nodes. The inter-satellite link refers to the link used for communication between satellites, which can reduce the number of inter-satellite hops and communication delay. In this kind of network, as the wireless coverage of nodes is limited, the data communication function is completed by single-hop communication or multi-hop forwarding. Within the maximum communication distance, network nodes communicate directly with neighbor nodes, and if the distance exceeds the maximum communication distance, routing technology is used to

find the optimal path for forwarding communication. The mainstream routing technology is distributed, and each communication node is used as a communication terminal and has the function of routing forwarding. In order to reduce the network transmission traffic and improve the system efficiency, the path with the shortest number of hops between the two nodes will be selected as the optimal path.

Literature [1] divides modern satellite routing algorithms into three strategies: centralized, distributed and hybrid. Early satellite networks used connection-based centralized routing algorithms. Ref. [2] proposed a finite state machine (Finite State Automata, FSA) routing algorithm which discretized and modeled the dynamic topology of the satellite network as a finite state machine, and allocated different routing tables for different network topologies in different states. However, it ignores the dynamic variability of inter-satellite links. The Flow Deviation (FD) proposed in [3] aims to find the path with the smallest end-to-end delay, and it is suitable for inter-satellite links with a small number of nodes. However, it is easy to cause packet loss and delay problems because static routing algorithms cannot perceive link state information. The proposed QoS routing algorithm MPQR in [4] uses a genetic algorithm and a simulated annealing algorithm at the source node to find the optimal path that satisfies the constraints of Quality of Service (QoS), which can simultaneously improve the end-to-end network delay, packet loss rate and link utilization. Different from centralized routing, each satellite node acts as an independent routing decision in distributed routing strategy. The Delay-oriented Adaptive Routing Algorithm (DOAR) proposed in [5] is used to solve the multipath load balancing and delay problems, while it is only for delay-sensitive services. Ref. [6] proposed Service-Oriented Routing with Markov Spacetime Graph (SOR-MSG) based on Markov Spacetime Graph. First, an MSG model is constructed through the connection duration and connection probability between satellite nodes. The satellites in the link autonomously select the next hop node based on the established MSG and QoS constraints, and finally determine the optimal path, but ignore the stability and reliability of the system. Hybrid satellite routing combines the characteristics of centralized routing and distributed routing. One of the ideas is to adopt a semi-distributed routing strategy, that is, there are nodes in the forwarding path that can independently determine the next hop node and nodes that are only responsible for transparent forwarding node. Ref. [7] proposed a weighted semi-distributed routing algorithm (Weighted Semi-Distributed Routing Algorithm, WSDRA). WSDRA divides satellite nodes into Routing Satellite (RS) and Messenger Satellite (MS). RS is responsible for routing path selection. It determines the next hop and the hop after the next for the packet forwarding. MS only transmits the packet according to the two-hop routing information carried by the packet. Hybrid routing combines the characteristics of centralized routing and distributed routing algorithms. Although it can improve the shortcomings of centralized routing and distributed routing, the algorithm has a high degree of complexity and is difficult to implement.

Nowadays, there are some articles on the research of satellite routing algorithms based on machine learning. In the intelligent routing algorithm of LEO satellite network based on reinforcement learning proposed by [8], the satellite node can adaptively select the next hop according to the reinforcement learning model, which has better delay characteristics than the traditional distributed routing algorithm. Ref. [9] combines Software-Design Networking (SDN) with deep reinforcement learning to solve the real-time optimization problem of the network, and the network delay is significantly reduced. Ref. [10] proposed an integrated adaptive routing algorithm ISTNQR (Integrated Satellite-Terrestrial Information Network based Q-Learning Routing Algorithm) based on the machine learning algorithm branch Q-Learning. With the help of the characteristics of the centralized control network of the controller under the SDN architecture, the disadvantage of the traditional Q-Learning algorithm that the distributed computing achieves long routing convergence time is improved.

In order to improve the reliability of inter-satellite network transmission and reduce congestion, this paper firstly proposes an inter-satellite network multi-path routing al-

gorithm based on load balancing. First, use the BFS (Breadth First Search) algorithm to search for all nodes with a distance of k from the source node, and then search for other nodes with a distance of k + l from the source node. According to the inter-satellite link load, the link with the least data occupancy of neighbor nodes and the lowest load of the entire path is selected as the transmission link from all the shortest paths to the destination node. In addition, in order to adapt to the dynamic nature of the network, when some network topology information cannot be achieved, an inter-satellite network dynamic routing algorithm based on reinforcement learning is proposed. The simulation results show the effectiveness of the algorithm.

## 2. Multipath Routing Algorithm for Inter-Satellite Links Based on Load Balancing

### 2.1. Inter-Satellite Network Topology

The network model examined in this research, seen in Figure 1, is an inter-satellite network made up of N satellite nodes, expressed as $G = \{[V_i, V_j], \ldots\}$. $V_i$, $V_j$ are neighbor nodes. Define $AdjMatr[i][j]$ as an adjacency matrix in which the subscript i and j map the node address and if a link exists between the nodes, the element value is 1; otherwise, it is 0.

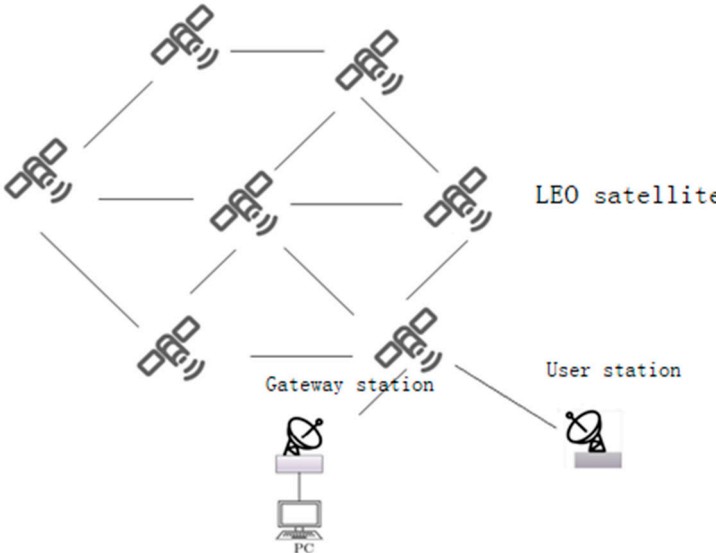

**Figure 1.** Schematic diagram of inter-satellite link.

The inter-satellite network nodes update the topology information of the entire network periodically, and calculate the current link adjacency matrix according to the link connection status between nodes in the entire network. We define $AdjMatr[i][j]$ to show the link connection status between node i and j. If there is a link between any two mobile nodes, it means that the two nodes are neighbor nodes, and the two-nodes adjacent matrix value is marked as m, otherwise the value is n.

$$AdjMatr[i][j] = \begin{bmatrix} n & m & \cdots & n & \cdots & n & n \\ m & n & \cdots & m & \cdots & n & n \\ \vdots & \vdots & \ddots & \vdots & \ddots & \vdots & \vdots \\ n & n & \cdots & m & \cdots & n & n \\ \vdots & \vdots & \ddots & \vdots & \ddots & \vdots & \vdots \\ n & n & \cdots & n & \cdots & n & m \\ n & n & \cdots & n & \cdots & m & n \end{bmatrix} \quad (1)$$

### 2.2. Multipath Routing Algorithm for Inter-Satellite Network

All of the neighbor nodes of the head node are sequentially added to the queue based on the adjacency matrix data of the nodes in the intersatellite network. Calculate the shortest paths between each neighbor node and record front nodes of the source node while traversing the neighbor nodes. The head node of the team will be dequeued when all the neighbor nodes have been traversed. Until the queue is empty, keep creating new queue head nodes sequentially. Finally, we can obtain the shortest paths and front node information between the source node to all nodes in the whole network.

*HeadAddr* is the address of the head node of the team, *AdjAddr* is the address of the neighbor node, and the variable *Enqueue* records whether the node in the network has joined the team. The variable *Dequeue* records whether the node in the network has been dequeued, and if it is true, it indicates that the node has been dequeued. *Queue* is used to store the queue of neighbor nodes. *HopCount[i]* is used to count the shortest hops from the node i to the source node. *FrontPoint[i]* is used to store the front node of the destination node while *FrontCount* store the total number of front nodes of the node.

Figure 2 is the algorithm flow chart, and the technical process is realized as follows:

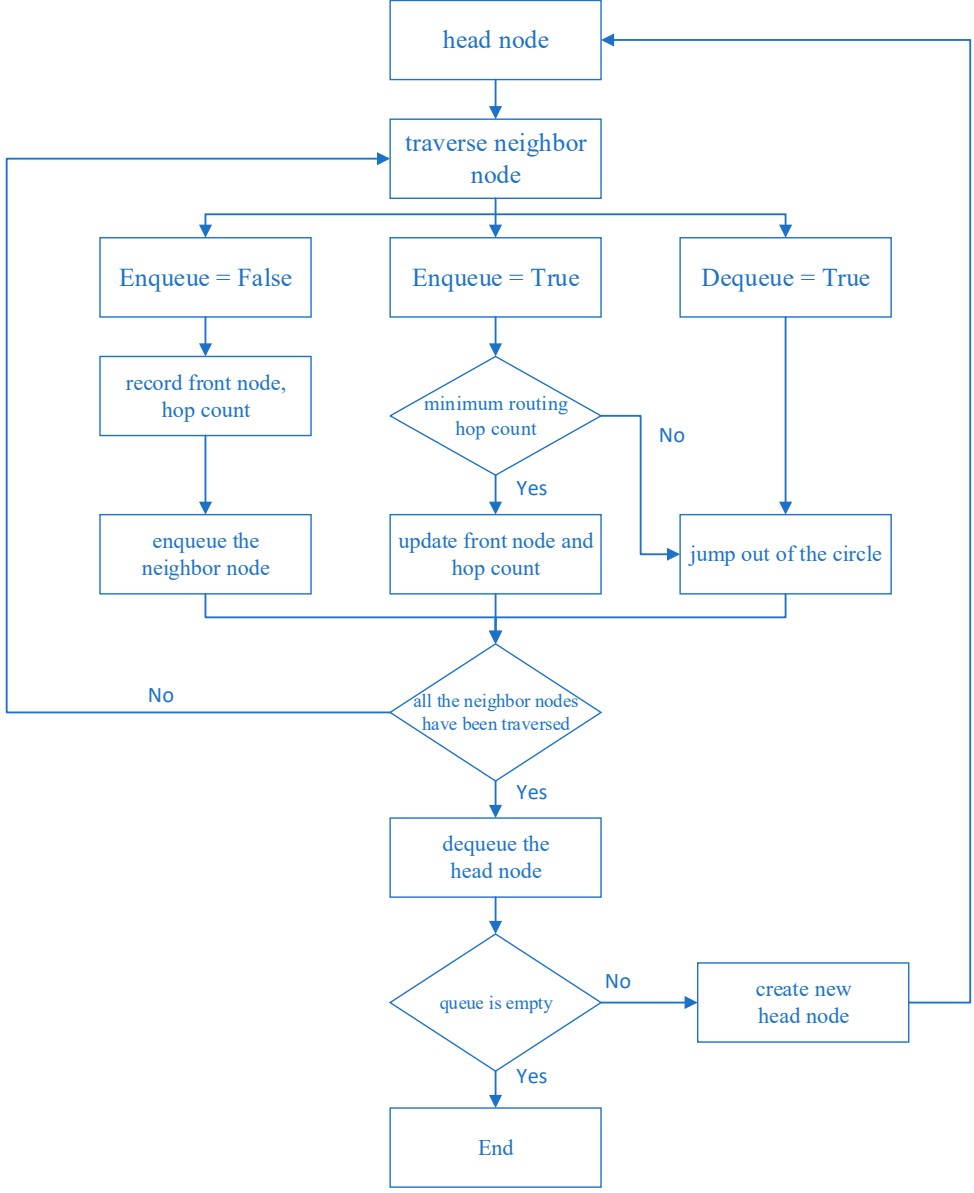

**Figure 2.** Flow chart of multipath algorithm.

**Step 1:** When the algorithm is started, first put the source node into the queue as the head node and then initialize the number of hops away from the source node *HopCount[HeadAddr]* as 0, traverse all the neighbor nodes which are represented by *AdjMatr*[*i*][*j*] = 1, and set the network address of one hop neighbor nodes as *AdjAddr*;

**Step 2:** If the *Enqueue* of the *AdjAddr* is *False*, then *HopCount[AdjAddr]* of the shortest hops between the neighbor node and the source node equal *HopCount[HeadAddr]* add 1. Secondly, put the head node of the queue traversed this time as the front node into the front node container *FrontPoint[i] = HeadAddr*, i = 0, 1, . . . n, and add 1 to the front node counter *FrontCount*. Finally, put the neighbor node into the queue and mark the *Enqueue* of the neighbor node as true;

**Step 3:** If the *Enqueue* of the neighbor node is *True*, which means that the neighbor node is also a neighbor node of other nodes and has been enqueued, then compare whether the shortest hop count of this cycle is the minimum value or not. If the shortest hops calculated by this loop are the smallest, indicating that this link is the optimal path, then update the shortest distance of the neighbor node to the shortest distance calculated this time, and update the front node to the current head node, keep the *Frontcount* unchanged; otherwise, the path obtained in this cycle is not optimal, and jump out of this cycle directly;

**Step 4:** If the *Dequeue* of the neighbor node is *True*, it means that the neighbor node has been dequeued. In order to avoid double counting, no information will be recorded for the dequeued nodes until all neighbor nodes have been traversed, and setting the *Dequeue* of the head node at the team is *True*.

At this point, the first round of traversal process of the head node of the queue has ended, the current first node of the team will be dequeued, and the next new head node of the team will be generated from the queue in turn. The above operations will be repeated until the number of nodes in the queue is empty, in the end the shortest paths from the source node to all other nodes and the front nodes of all other nodes have been obtained.

*2.3. Multipath Backtracking*

All the shortest pathways from the source node to other nodes are traced layer by layer in accordance with the front nodes, starting with the source node as the root node. To determine the number of hops from every node in the entire network to the source node, follow the steps in Figure 3: first, trace back the path of the one-hop node, then trace back the path of the two-hop node, until the shortest path of the farthest node in the network is determined.

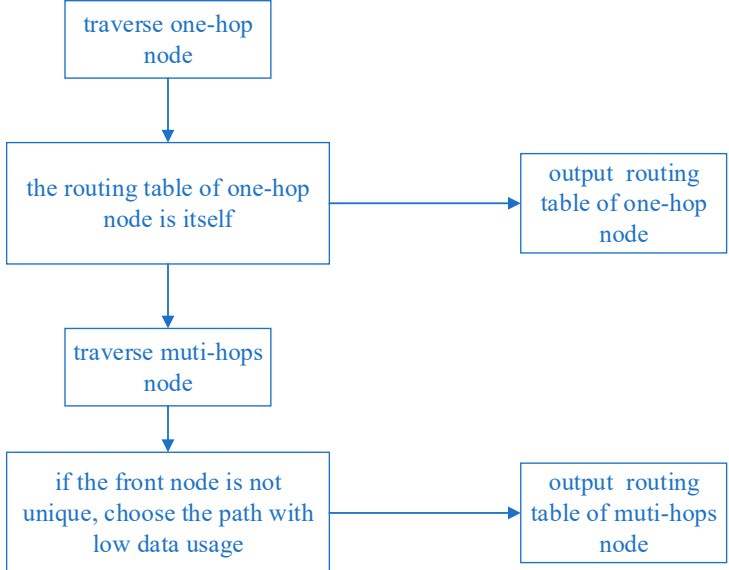

**Figure 3.** Multipath backtracking.

*2.4. Optimal Path Selection for Inter-Satellite Links*

After obtaining multiple paths, if there is only one shortest path, this path is selected for data transmission. If there are multiple shortest paths, the entire path and the path with the largest bandwidth capacity of neighboring nodes are selected first from all the shortest paths, which can avoid using a highly congested path to improve the system transmission throughput and balance the load. Choosing neighbor nodes with less data usage can avoid the situation where a complete path has the least load, but a single node has a very high data usage rate or even reaches the threshold, and the data usage rate of other nodes on the path is extremely low. If it only relies on the variable with the lowest occupancy rate of all data on the path, data congestion is very likely to occur. In order to avoid this situation as much as possible, we consider the neighbor nodes of the source node as the target, and according to experience, the neighbor nodes, as the first hop node for the source node to transmit services to the distant destination node, often carry a relatively important role. Especially when the source node has business transmission requirements to multiple long-distance destination nodes, if there are multiple neighbor nodes, choosing a suitable neighbor node can not only quickly complete the business needs of each destination node, balance the load, improve the throughput of the entire system, but also realize shunting, which can avoid multiple business needs choosing the same neighboring nodes, affecting the transmission requirements of multiple services, the throughput of the entire system and the delay.

Use s to record the number of transmission connections of inter-satellite network nodes, w is the weight to represent the node bandwidth capacity, the greater the node bandwidth capacity, the greater the corresponding weight. When there is a data service demand, the link data usage value of the node is increased by 1. If the node fault or is congested, the node weight is set to 0, and the node is no longer used for data transmission. Assuming that the total number of nodes in the network is m, the number of shortest paths obtained is n, and a link with the smallest ratio of node connection number to weight is selected from n links as the optimal path. Use c to represent the number of links, w to represent the weight, s to represent the selected node, and sn to represent the unselected node. We use $c(s)/w(s)$ to measure the link state and then the selected node link S must satisfy: $c(s)/w(s) < c(sn)/w(sn)$.

## 3. Dynamic Routing Algorithm for Inter-Satellite Network Based on Q-Learning

The multi-path routing technique suggested in the second part is no longer appropriate since many satellites are constantly moving at a rapid speed and occasionally it is impossible to obtain the global topology of the inter-satellite network. There are many studies on service networks using machine learning and reinforcement learning, including service category, routing decision, service function chain scheduling and resource optimization, etc. [11–17]. This section suggests a Q-learning-based dynamic routing technique for inter-satellite networks as a result.

*3.1. Q-Learning*

In Q-learning, which belongs to the field of reinforcement learning, agents/decision makers try to interact with the environment by learning the behavior of dynamic systems. In this chapter, the agent receives the current state and rewards of the dynamic system, and then performs corresponding actions based on its experience to increase long-term revenue through state transitions. States and rewards represent data the agent receives from the system, while actions are the only input to the system. Unlike supervised learning, in Q-learning, the agent must find the optimal action to maximize the reward while the agent's actions not only affect the current reward, but also affect future rewards.

A trade-off exists in Q-learning between exploitation and exploration. Unknown actions are investigated in order to avoid missing better candidate actions, but because they are unpredictable, they may worsen network performance. On the other hand, if action is

dependent on the present best action choice, it may result in a local optimal solution even when other as yet undiscovered actions might offer more advantages.

Figure 4 describes the basic Q-learning algorithm flow, and its specific execution steps are described as follows:

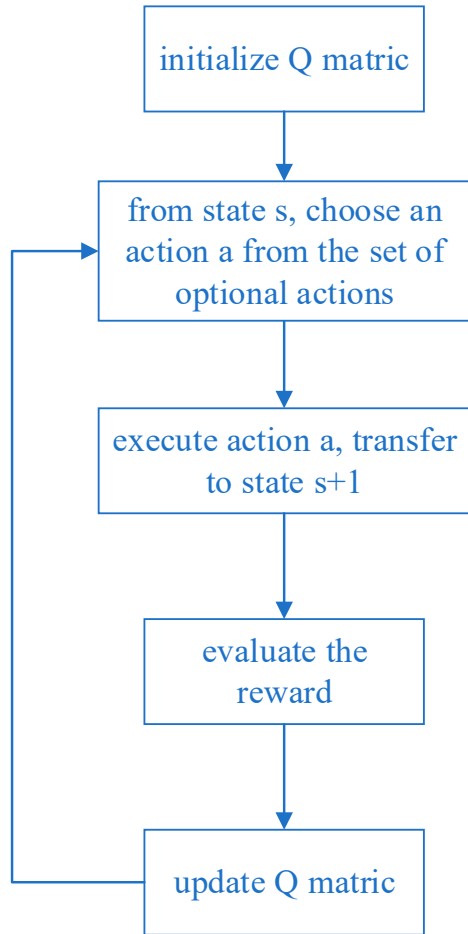

**Figure 4.** Q-learning algorithm flow chart.

**Step 1**: Initialize the Q value. This stage constructs a Q-table, where the rows represent the state space and the columns represent the action space, and are initialized to 0.

**Step 2**: Throughout the lifetime, repeat Step 3–Step 5 until the set number of stop training is reached.

**Step 3**: Based on the current Q value estimation state, select an action from the optional action space, which involves the trade-off between exploration and exploitation in Q-learning. The most commonly used strategy, for example, $\varepsilon - greedy$, adopts an exploration rate as the random step size. It can be set larger initially, because the Q value in the Q-table is unknown, and a large amount of exploration needs to be performed by randomly selecting actions. Generate a random number and if it is greater than $\varepsilon$, the action will be selected using known information, otherwise, it will continue to explore. As the agent becomes more confident in the estimated Q value, $\varepsilon$ can be gradually reduced.

**Step 4–5**: Take action $\alpha$ and observe the output state, evaluate the reward, then use the Bellman equation to update the $Q(s, a)$ function.

### 3.2. Q-Learning Routing System Model

Various services in the satellite network have specific requirements, such as bandwidth, jitter, delay, etc. In order to better meet the QoS requirements of diverse services, intelligent path selection and traffic distribution are crucial. There are multiple optional

paths between the source node and the destination node based on service requests, and each path in the network may have different bandwidth and inherent delay. Therefore, the introduction of an SDN controller can intelligently provide a suitable transmission path for each service flow.

Figure 5 shows the structural framework of SDN based on Q-learning. The SDN control plane contains the intelligent decision-making module of Q-learning. The intelligent decision-making module effectively generates network policies and realizes global, real-time and customizable network control and management. Specifically, when a service request arrives, the SDN controller obtains the global network state, and according to the link state in the current network, coupled with Q-learning action learning, constantly explores various optional paths. Therefore, the intelligent decision-making module generates the optimal path decision, and sends the forwarding rules to the nodes. At this time, the node can forward the data packet according to the routing table. Through this architecture, routing can be intelligently and rationally selected according to network resources to improve network performance.

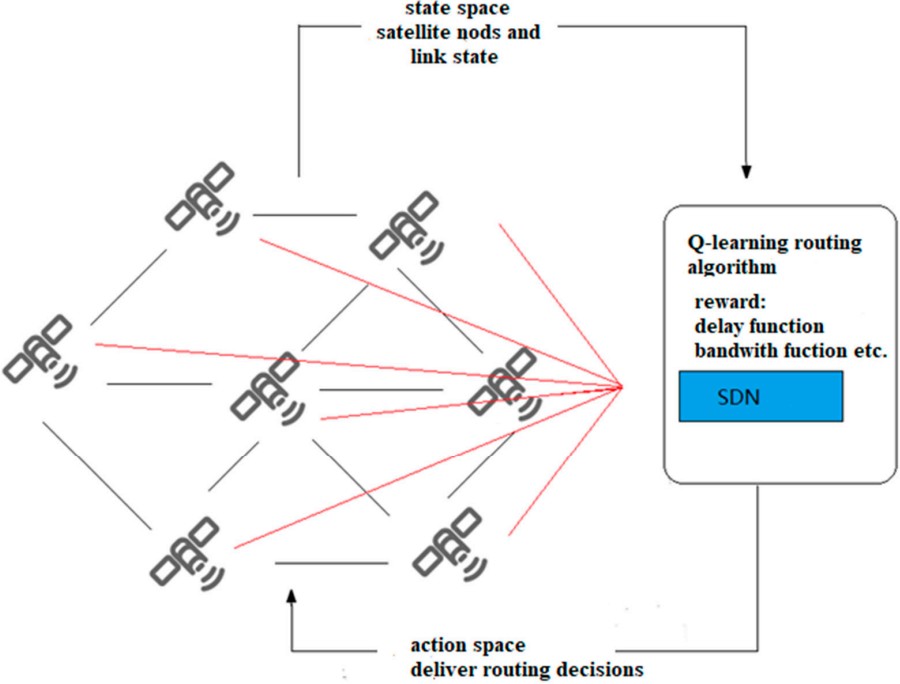

**Figure 5.** SDN routing framework based on Q-learning.

In this section, the inter-satellite routing model in the SDN network is established to provide the best transmission path for business flows, meet its QoS requirements, and reduce the utilization rate of the link with the highest load and transmission delay in the network as much as possible.

### 3.2.1. Network Model

The network data plane in SDN is represented as an undirected graph of n nodes, each node represents a satellite, and n nodes are connected by undirected links. The network topology is represented by a graph G = (V, E), where V represents a set of nodes with |V| = n, and E represents a set of links connecting nodes in the network. Assuming that G is a connected graph without any isolated nodes, the link bandwidth capacity, transmission delay and packet loss rate are expressed as $b_{ij}$, $d_{ij}$ and $l_{ij}$.

### 3.2.2. Business Characteristics

The business $S^k$ mainly includes the following characteristics, the source node $s^k$ (the node entering the network), the destination node $t^k$ (the node leaving the network), the

bandwidth requirement is set to $b^k$, other QoS requirements include transmission delay and packet loss rate, which are represented by $D^k$ and $L^k$, respectively, referring to the highest acceptable delay and packet loss thresholds for this service.

Let $x_{ij}^k$ denote whether the SDN controller assigns the link to transmit the service flow, if the service flow passes through the link, then set $x_{ij}^k$ to 1, otherwise it is 0.

The optimization goal is to balance the link load in the network and reduce the transmission delay as much as possible on the basis of ensuring the QoS requirements, that is, to minimize the Formula (2a).

$$\lambda \max_{(i,j)\in E} load_{ij} + (1-\lambda) \sum_{k\in K} \sum_{(i,j)\in E} x_{ij}^k \cdot d_{ij} \tag{2a}$$

where $load_{ij}$ is the bandwidth utilization of the link $(i,j)$, $load_{ij} = \frac{\sum_{k\in K} x_{ij}^k * b^k}{b_{ij}} \times 100\%$, $\sum_{(i,j)\in E} x_{ij}^k \cdot d_{ij}$ denotes the cumulative delay of the selected path, parameter $\lambda \in [0,1]$ used to adjust the proportion of link load and delay.

The main constraints under this model are as follows:

$$\sum_{(i,j)\in E} x_{ij}^k - \sum_{(j,i)\in E} x_{ji}^k = \begin{cases} 1, & i = s^k \\ 0, & i \neq s^k, t^k \\ -1, & i = t^k \end{cases} \tag{2b}$$

$$x_{ij}^k \cdot b^k \leq \min(b_{ij} - \sum_{k_0 \in K, k_0 \neq k} x_{ij}^{k_0} \cdot b^{k_0}), \forall k \in K \tag{2c}$$

$$\sum_{(i,j)\in E} \sum_{k\in K} x_{ij}^k \cdot b^k \leq b_{ij} \tag{2d}$$

$$\sum_{(i,j)\in E} x_{ij}^k \cdot d_{ij} \leq D^k, \forall k \in K$$
$$\sum_{(i,j)\in E} x_{ij}^k \cdot l_{ij} \leq L^k, \forall k \in K \tag{2e}$$

Constraint (2b) represents the flow conservation in the traffic network link, Formula (2c) means that the current available bandwidth of the link meets the transmission rate requirements of its business flow, and constraint Formula (2d) means that the traffic carried by the link does not exceed its link bandwidth capacity. The following constraint expression shows that in addition to meeting the transmission rate requirements, other QoS requirements of the business must also be met. Based on the routing framework of SDN, the Q-learning algorithm is further introduced into the routing problem, the reward function is designed, and a dynamic routing algorithm based on Q-learning is proposed.

*3.3. Dynamic Routing Algorithm Based on Q-Learning*

In this section, a Markov decision process is established for the path selection problem of the inter-satellite link. The arrival and departure of business flows are regarded as random processes, and different types of business flows have different statistical characteristics. We assume that business flow requests arrive independently. Furthermore, the probability of the business type follows a pre-specified Poisson distribution. The SDN controller must make a decision at each discrete time interval to accept or reject new service requests based on the current system state. In the case of acceptance, it is decided to allocate the best transmission path for the traffic flow. If declined, there is no need to allocate resources for it.

The Markov decision process (MDP) provides a mathematical framework for modeling Q-learning systems, represented by a quadruple $(S, A, P, R)$, where $S$ represents a finite state set, $A$ a finite action set, $P$ the state transition probability, and R a reward set. The relationship among state, action and reward function can be expressed by Equation (2f). The state transition function: $S \times A \to P(s)$, where $P(s)$ is the probability distribution for state $s$, and the reward function returns a reward $R(s,a)$ after a given action $a \in A(s)$, where $A(s)$ is the set of available actions for state $s$. The learning task in MDP is to find the policy $\pi$: maximizing the cumulative reward. To maximize the reward received across

all interactions, the agent must choose each action according to a strategy that balances exploration (acquisition of knowledge) and exploitation (use of knowledge). Store the value function (or Q value) of each state-action pair in the MDP. $Q(s, a)$ is the expected discounted reward when executing action $a$ in state $s$ following policy $\pi$.

$$S \times A \rightarrow R \tag{2f}$$

In addition to agents/decision makers and dynamic systems, Q-learning also includes decision functions, value functions, long-term rewards, etc. [18]. The decision function refers to the policy that the agent will take, which maps from the perceived system state to the corresponding action, guiding the action of Q-learning. The exploration strategy we use in this chapter is $\varepsilon - decreasing$, where the probability of choosing a random action (exploration) is $\varepsilon \in [0, 1]$ and choosing the best action (exploitation) is $1 - \varepsilon$. In this way, at the beginning, maintain a high exploration rate, and each subsequent episode $\tau \in \Lambda$ will decrease according to $\varepsilon_\tau = \sqrt{1 - [\tau/(4 \times |\Lambda|)]^2}$.

The state-action value function characterizes the value of a state-action pair, indicating the difference between the current state and the stable state. The Q value function is updated as shown in Formula (2g).

$$Q(s_t, a_t) = (1 - \alpha)Q(s_t, a_t) + \alpha \left[ R(s_t, a_t) + \gamma \max_a Q(s_{t+1}, a_{t+1}) \right] \tag{2g}$$

$\alpha \in [0, 1)$ is the learning factor of Q-learning, which represents the rate of the newly acquired training information covering the previous training, and $R(s_t, a_t)$ is the reward at time t. $\gamma \in [0, 1]$ is the discount factor that determines the importance of future rewards. In Equation (2g), specifically, at time t, the agent performs action $a_t$ on the current state $s_t$ of the routing optimization model and then reaches state $s_{t+1}$, and at the same time, the feedback loop will report the reward function $R(s_t, a_t)$ to the agent, and update the action value function $Q(s_t, a_t)$ and Q matrix accordingly. Then, the agent repeats the above operation for the state $s_{t+1}$, and so on until the optimal action value $Q^*(s_t, a_t)$ is reached, the agent selects the optimal strategy $\pi_Q{}^*$ according to the rewards of each strategy in $Q^*(s_t, a_t)$, and the optimal strategy can be expressed as Formula (2h);

$$Q^*(s_t, a_t) = E \left[ R(s_t, a_t) + \gamma \max_{a \in A} Q^*(s_{t+1}, a_{t+1}) \right] \tag{2h}$$

Long-term rewards indicate the total reward an agent can expect to accumulate over time for each system state.

The key to planning a path for a service flow is to select an appropriate path for packet forwarding according to the service requirements and the real-time status of each link in the network. The key to flexible selection of the best transmission path is that there are multiple optional paths in the network, and the states of links are different. In order to avoid data traffic congestion during inter-satellite network communication, data packets re-plan another optional path. This optional path may not be the shortest, but its link utilization rate is low, while for delay-sensitive services, the impact of link delay cannot be ignored, so the reward function is set as an index of comprehensive link states as shown in Formula (2i), in which $\alpha_1, \alpha_2, \alpha_3$ are the adjustment factors of link bandwidth, delay and packet loss rate respectively.

In this chapter, we divide the link bandwidth into three levels, the link load above 80% is regarded as excessive use, the link between 30% and 80% is regarded as moderate load, and the rest is less than 30% as the link load is light. In this way, the state set is roughly classified, but considering that there are differences in link utilization at each level, for example, it is obviously unreasonable to directly regard 40% and 70% link bandwidth utilization as the same level, so when designing the bandwidth part of the reward function R of Q-learning, the link load of the moderate level ([30–80%]) is divided more finely to reflect the difference.

According to the above analysis, the bandwidth part of the reward function is set based on the sigmoid function, as shown in Figure 6. The abscissa represents the remaining link bandwidth ratio (inversely proportional to the link load), while the ordinate represents the immediate reward of the link bandwidth part $delay_{ij}$. It is characterized in that when the link utilization rate is overloaded, the reward value of the bandwidth part tends to almost 0, and the mid-term transition is smooth, but the gradient value is getting higher and higher, until the link is idle (occupancy rate is lower than 30%), its reward value approaches 100. $delay_{ij}$ and $loss_{ij}$ represent the ratio of the delay and packet loss rate of link $(i, j)$ to the maximum delay and packet loss in the global link, respectively, thus, the values are all between (0,100]. Finally, the reward function is shown in formula (2i). Specifically, if the selected link is in a better state, the reward value will be greater after the agent acts.

$$R_{i \to j} = R\left( i, j|_{s_t, a_t} \right) = -\cos t + \alpha_1 BW_{ij} - \alpha_2 delay_{ij} - \alpha_3 loss_{ij} \qquad (2i)$$

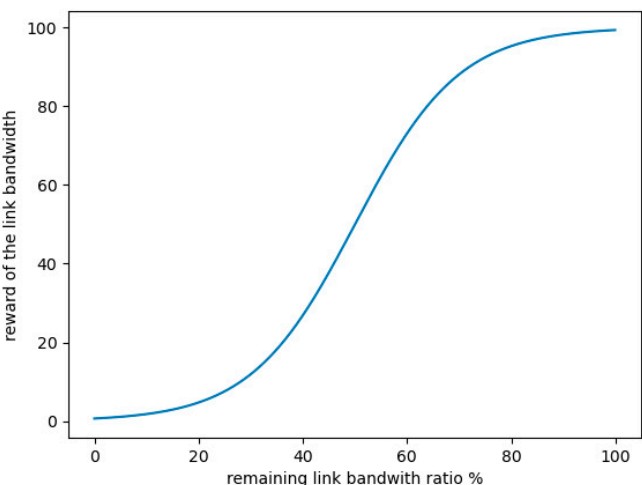

**Figure 6.** Reward function curve based on link bandwidth remaining.

In this chapter, the routing system based on Q-learning is composed of SDN controller and satellite nodes. The SDN controller acts as an agent to interact with the environment and obtain three signals: state, action and reward. Among them, the state space is represented by the traffic matrix of the nodes and the links between the nodes, which represents the current network link traffic load. The agent takes which node to forward the data packet to as an action space. The reward function is related to the type of business. If a business is delay-sensitive, the reward function will increase the weight of the delay part, so that the agent can change the path of the data flow, and the corresponding flow table will be sent to the corresponding nodes. The reward function R for the agent to perform an action is related to the operation and maintenance strategy of the satellite network. It can be a single performance parameter, such as delay, throughput, or comprehensive parameters [19].

The routing algorithm proposed in this chapter is based on the classic reinforcement learning technique Q-learning for finding the optimal state-action policy for MDP. Under certain conditions, Q-learning has been shown to be optimal [20]. On the other hand, in the Q-routing algorithm, the decision strategy $\varepsilon-$decreasing is a reasonable exploration and utilization strategy, which also meets the requirement of optimality, so the Q-routing algorithm proposed in this chapter can converge and approach the optimal solution. Regarding the convergence efficiency, it has been proved in [21] that Q-learning will sufficiently converge to the $\xi$ neighborhood of the optimal value after $(N \log(1/\xi)/\xi^2)(\log N + loglog(1/\xi))$ iterations, where N represents the number of states in the MDP, which also ensures that the algorithm proposed in this chapter can converge in a finite number of steps.

In this chapter, the routing problem is reduced to let the agent learn a path from the source node to the destination node. In the routing problem, according to the established

strategy $\pi$, the agent can get a path connecting the source node to the destination node. Next, we introduce the Q-routing algorithm proposed in this chapter in detail.

When the SDN controller plans the transmission path for the business flow, it will search the global space to obtain the best action. The MDP model is as follows: any switch node $v \in V$ in the SDN network can be regarded as a state $s$. Each state $s \in S$ has an optional action set $A(s)$, which is composed of links connecting the state of $s$, and the reward function is given in the previous section.

The SDN network contains many loops. In order to reduce the search step of the algorithm from the source node to the destination node as much as possible, the maximum execution time of the agent's action is set to $t_{max}$. Therefore, the SDN controller is required to plan an optimal path for the service flow within a time interval $t_{max}$, otherwise the path allocation is terminated. Although the termination search strategy belongs to the invalid strategy set $\prod_{useless}$, it is still used as an option for the agent (SDN controller) to search for actions. On this basis, the feasible strategy set under the algorithm is defined as $\prod_{useful} = \left\{ \pi_{i,j}^{s,t}(t_i) \right\} \Big| i = 1, 2, \ldots, M; j = 1, \ldots, N_i$, where M represents the current time interval, and $N_i$ indicates the total amount of policies from the source node to the destination node. Correspondingly, the invalid strategy set is defined as $\prod_{useless} = \left\{ \pi_{i,j}^{s,V_{t_i}}(t_i') \right\} \Big| i = 1, 2, \ldots, M'; j = 1, \ldots, N_i'$, where $M'$ represents the current time interval, $N_i'$ is the sum of all strategies from source node to node $V_{t_i}$. Then the set of all possible strategies is $\prod_{(s,t,t_{max})} = \prod_{useless} \prod_{useful}$. Algorithm 1 gives the pseudo cod e of this algorithm.

---

**Algorithm 1** Dynamic Routing Algorithm Based on Q-learning (Q-routing)

---

**Input**: Network topology information $G = (V, E)$, Business request information
**Output**: Q matric, Path policy

1.    Initialize

2.    learning factor $\boldsymbol{\alpha}$, discount factor $\gamma$, maximum execution time $\boldsymbol{t_{max}}$,

set Q matric to 0, reward $R_0$

3.    for $S^k$ in business set S

4.    while $t \leq \boldsymbol{t_{max}}$

5.    current state $\boldsymbol{s_t}$

6.    for all optional actions $\boldsymbol{a_t}$ in the state $\boldsymbol{s_t}$, form the action set $\boldsymbol{A_c}$;

7.    while $A_c \neq 0$

8.    choose an action $\boldsymbol{a} \in \boldsymbol{A_c}$ based on $\boldsymbol{\varepsilon - decreasing}$, shift to next state $\boldsymbol{s_{t+1}}$

9.    calculate the reward $\boldsymbol{R_t}$ and feedback it to the agent

10.   if state $\boldsymbol{s_{t+1}}$ == Goal state

11.   $\boldsymbol{R_t = R_t(s_t, a_t) + 100}$;

12.   else

13.   proceed to step 6 to select the action for the next moment $\boldsymbol{t + 1}$

14.   update Q matric based on the formula 4.8

15.   end if

16.   end while

17.   t = t + 1, update learning factor $\boldsymbol{\alpha_t}$

18.   end while

19.   end for

---

## 4. Simulation Experiment and Analysis

### 4.1. Performance Evaluation of Multipath Routing Algorithms

In order to verify that the multi-path routing algorithm of the inter-satellite link system proposed in this paper can realize shunting and improve system performance, we carried out simulation experiments on waiting delay, packet loss rate, and network throughput indicators. In the simulation experiment, the sending rate of business data packets increased from 10 packets/s to 100 packets/s, and the simulation time was set to 1 h. Taking the 16-node inter-satellite network shown in Figure 7 as an example, assuming the following topology, it is required to calculate the routing table from node V7 to other nodes, and select the optimal path according to the path load.

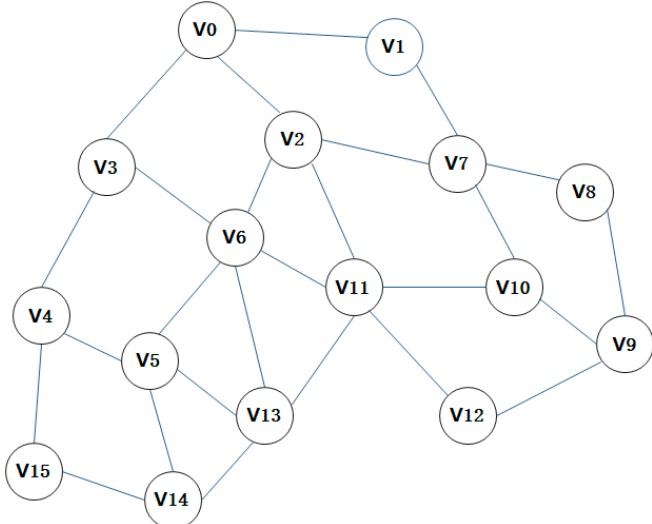

**Figure 7.** Simulate the network connection of inter-satellite nodes.

According to the above inter-satellite network connection conditions, a total of 16 paths from the source node V7 to destination nodes V0, V9, V11, V3, V12, and V13, were selected for service transmission testing. All the paths obtained by the algorithm in this paper are as follows: [V7->V1->V0], [V7->V2->V0], [V7->V8->V9], [V7->V10->V9], [V7->V2->V11], [V7->V10->V11], [V7->V1->V0->V3], [V7->V2->V0->V3], [V7->V2->V6->V3], [V7->V8->V9->V12], [V7->V10->V9->12], [V7->V2->V11->V12], [V7->V10->V11->V12], [V7->V2->V6->V13], [V7->V2->V11->V13], [V7->V10->V11->V13].

The paths obtained by the breadth-first search algorithm were [V7->V1->V0], [V7->V8->V9], [V7->V2->V11], [V7->V1->V0->V3], [V7->V8->V9->V12], [V7->V2->V6->V13]. There is only one path to the destination node V0, V9, V11, V3, V12, and V13.

We tested the business packets from node V7 to node V0, V9, V11, V3, V12, and V13, respectively, and recorded the arithmetic mean value of waiting delay, packet loss rate, and network throughput of each pair of nodes using traditional routing and the routing algorithm proposed in this paper.

### 4.1.1. Waiting Delay

With packet filling for a single path, as the service data packet sending rate increases, the available bandwidth margin becomes less and less, resulting in waiting delay conflicts. The experimental data in Figure 8 shows that the larger the data packet sending rate, the longer the waiting delay. When there is a demand for data package business, the algorithm first chooses a path with the least amount of business. As the rate of data package sending increases to a certain threshold, the algorithm chooses the path with less business consumption among the other shortest links as the transmission path to disperse the business packets to different neighbor nodes, decreasing the wait conflicts that occur as the rate of sending business packets increases.

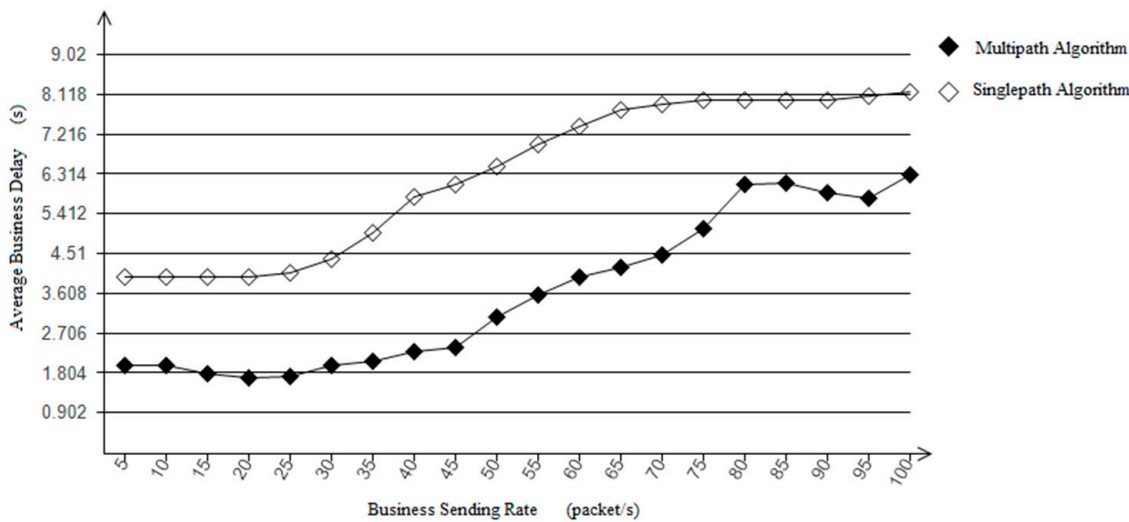

**Figure 8.** Average business wait delay comparison chart.

### 4.1.2. Packet Loss Rate

The simulation results in Figure 9 show that before the sending rate is less than 40 packets/s, both packet loss rates are extremely low because the amount of data does not reach the bandwidth capacity. With the increase in the service sending rate, the loss rate of both algorithms increases, but the loss rate of multipath algorithm is significantly lower than single-path algorithm, because the increase of data volume puts pressure on bandwidth capacity, and single-path transmission is difficult to guarantee the delivery of data and is prone to large-scale loss of packets.

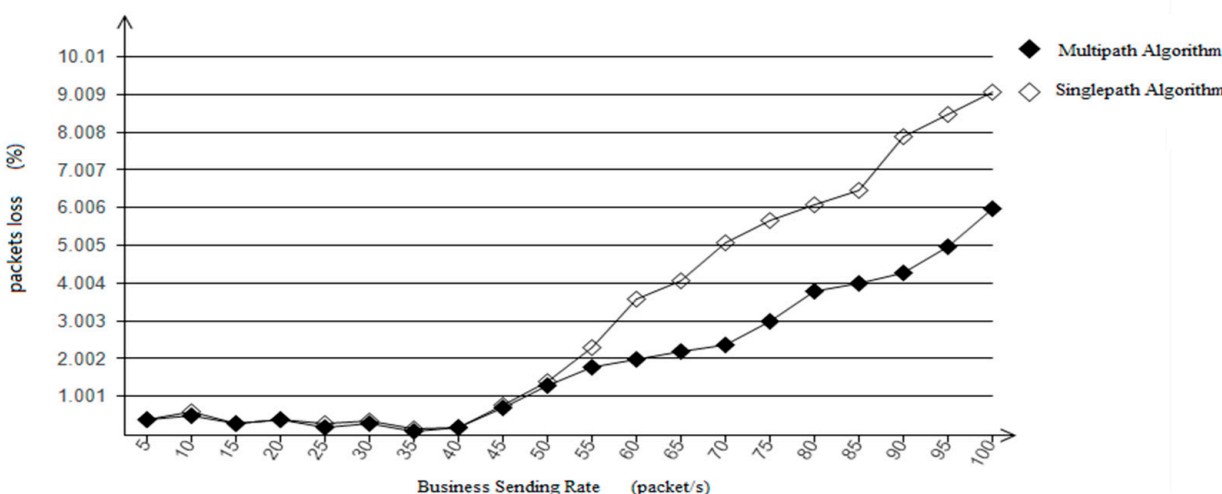

**Figure 9.** Packets loss comparison chart.

### 4.1.3. Network Throughput

The network throughput is measured by the service reception rate of the destination node. From the simulation data Figure 10, it can be seen that the throughput increases with the increase of the service packet transmission rate in both cases. However, the throughput performance of this algorithm proposed in this paper is significantly better than that of other algorithms.

It can be seen from the above that this algorithm can obtain all the shortest paths from source nodes to other nodes, avoiding the limitation of only one shortest path, and is able to choose the best path according to the load of neighboring nodes and their links. It can

improve link utilization, avoid data packet loss problems caused by link congestion or failure, and have a smaller wait delay and better network throughput performance.

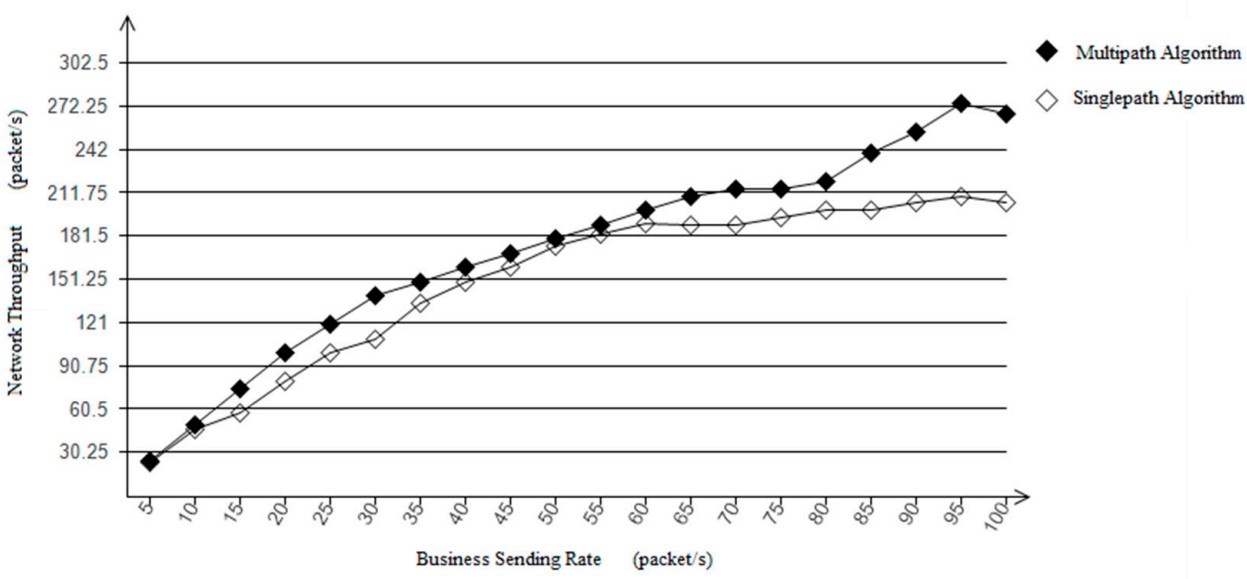

**Figure 10.** Network Throughput Comparison Chart.

*4.2. Q-Learning Routing Algorithm Performance Evaluation*

In addition, take the 16-node inter-satellite network topology shown in the previous section as an example. In order to comprehensively evaluate the validity of the model proposed in this paper and the feasibility of the algorithm, this chapter takes the load of the service transmission path and the average transmission delay of the service as the main performance index. This chapter compares the proposed Q-routing routing algorithm with Dijkstra's shortest path forwarding algorithm.

Firstly, the proposed Q-learning based routing algorithm is explored with learning factor $\alpha$ and discount factor $\gamma$. According to the Formula (2g), it can be seen that the larger the learning factor $\alpha$, the less the previous training results are retained, and the larger the discount factor $\gamma$, the greater the effect of considering future rewards. Figure 11a,b, respectively describe the comparison of Q value fluctuation (Euclidean distance) when $\alpha$ is set to 0.6 and 0.9, respectively, under different search steps with different discount factors.

The simulation results in Figure 11 show that when the $\alpha$ remains unchanged, the Q value fluctuation decays faster as the $\gamma$ increases. When $\alpha = 0.6$ and $\gamma$ is 0.3, it is about 30 steps that the Q value can be converged. When $\gamma$ is 0.6 and 0.9, the convergence speed is accelerated, but there are still different degrees of oscillation. When the learning factor increases to 0.9, it can be seen that the convergence speed of the Q matrix is significantly accelerated, and the fluctuation is smaller, and the effect is better.

However, speeding up the convergence rate of Q-routing routing needs further study. At the beginning of training, a larger learning factor can improve the convergence speed of Q matrix. However, as the training level increases, a larger learning factor causes the Q-matrix to move back and forth on both sides of the optimal point. Therefore, at the beginning of training, the learning factor can be set to a larger value, and with the increase of the number of iterations, the learning factor gradually decreases, that is, to achieve dynamic control of the learning factor. The learning factors are updated dynamically, and the update strategy is shown in Figure 12.

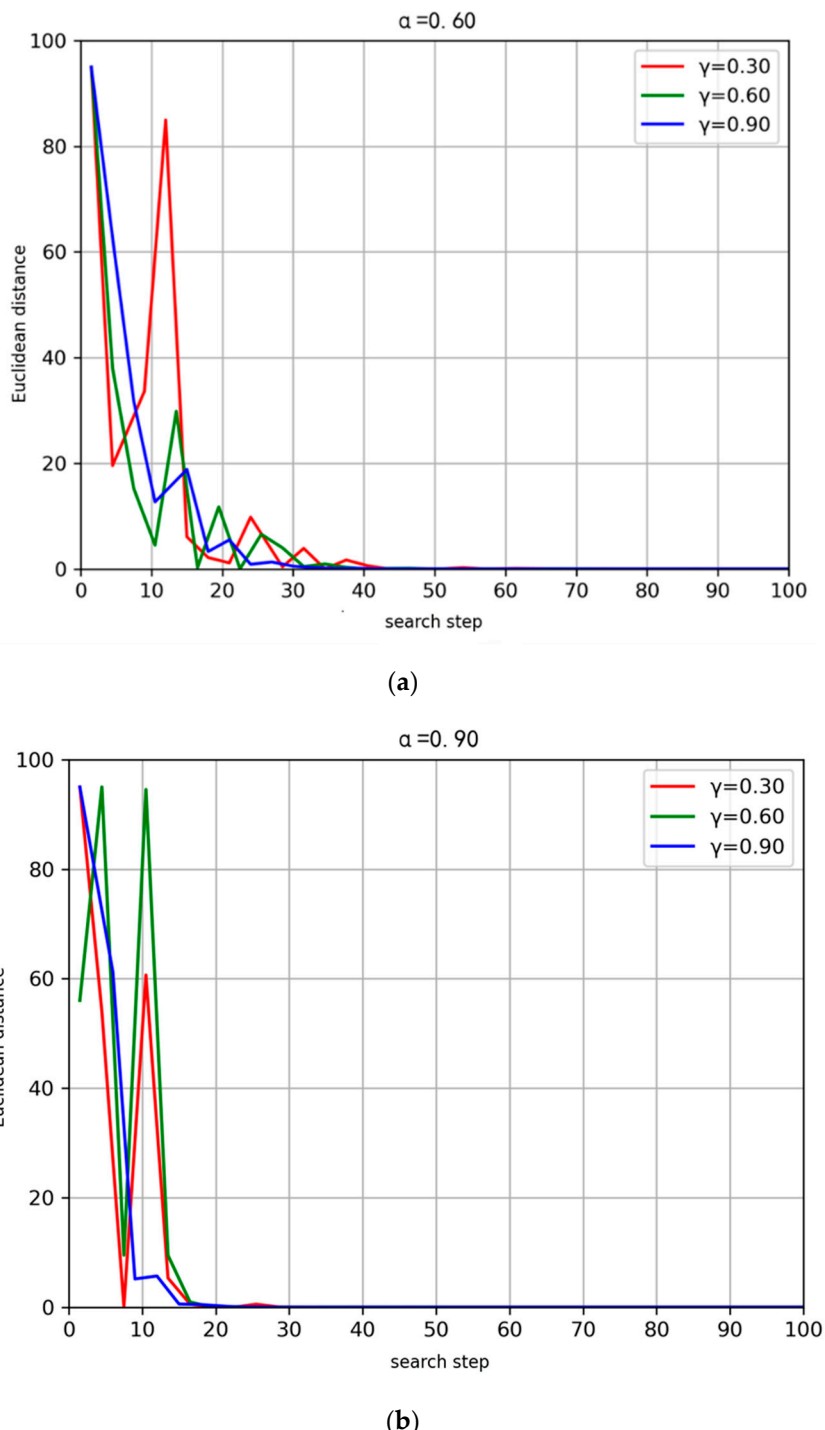

**Figure 11.** The comparison between the degree of convergence of the Q matrix and the number of search steps under different values of $\alpha$ and $\gamma$. (**a**) $\alpha = 0.6$, under different $\gamma$, the comparison chart of the convergence degree of Q matrix. (**b**) $\alpha = 0.9$, under different $\gamma$, the comparison chart of the convergence degree of Q matrix.

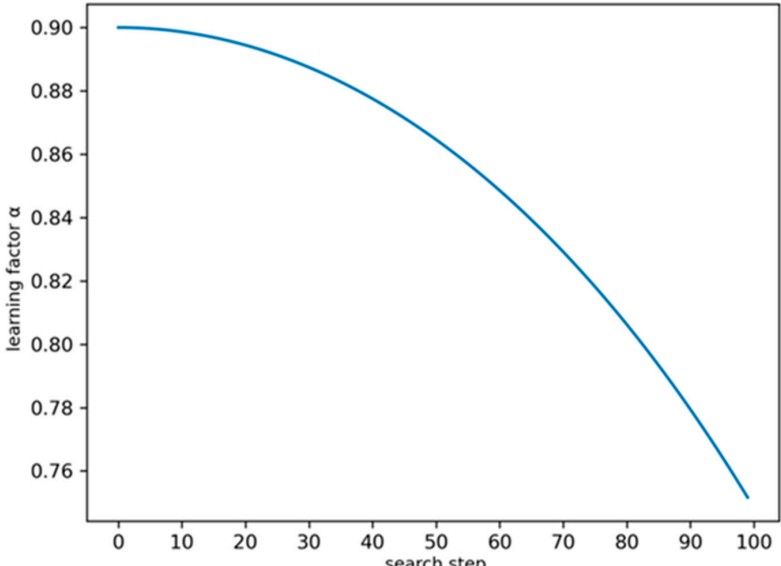

**Figure 12.** Learning Factors $\alpha$ Dynamic Change Curve with Training Step.

Figure 13 depicts the comparison of convergence before and after Q-routing algorithm updates the learning factor. The results show that after setting the dynamic learning factor, due to the initial $\alpha$ value is larger, the learning rate is faster, and then decreases gradually. A lot of previous training results are retained, that is, using the previous experience, so that the convergence rate of Q-matrix is improved to some extent.

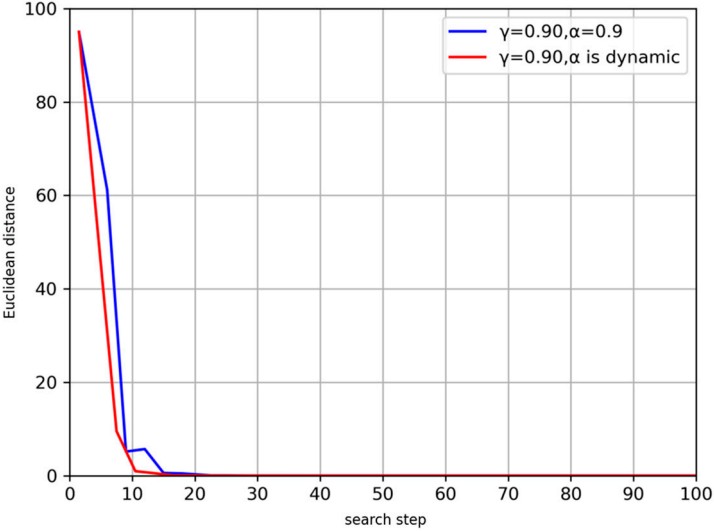

**Figure 13.** Q-matrix convergence before and after updating learning factors dynamically.

Next, we compare the Q-routing algorithm with the Dijkstra algorithm and the Q-learning algorithm proposed in [22]. The Q-learning routing algorithm can be regarded as a combination of the Dijkstra algorithm and the Q-learning algorithm. It predicts the link load and switches to Q-learning algorithm after reaching 80%. The reward function is relatively simple. When reaching the target node, it obtains 30 immediate rewards. When reaching the neighbor node of the target node, it obtains 20 rewards. This routing algorithm converges faster. It can also alleviate link congestion to some extent.

### 4.2.1. Path Load

Figure 14 depicts the maximum load of the selected business transmission path under different algorithms, where the reward function parameter in the Q-routing algorithm is $a_1$ for 0.8, $a_2$ for 0.2, $a_3$ for 0. Figure 14a indicates that services are generated at two fixed nodes in a satellite network, while Figure 14b indicates requests for services between random source and destination node pairs. As shown in Figure 14a,b, when the number of services is small, the Q-routing routing algorithm has the lowest load, while for the Q-learning algorithm and the Dijkstra algorithm, the link load increases rapidly. This is because the link load is added to the reward function of the Q-routing algorithm as an important parameter, which allows the agent to consider the link load when exploring the optimal path, thus optimizing the overall traffic balance. The Dijkstra algorithm prefers the shortest path for packet forwarding, and the shortest path increases the link load due to its frequent use. The performance of the Q-learning algorithm is consistent with that of the Dijkstra algorithm before the link load reaches the threshold of 80%. As the business intensity increases, the Q-learning algorithm searches for other alternative paths to avoid the shortest path, so the link load it chooses later is between the Q-routing algorithm and the Dijkstra algorithm. From Figure 14, it can be seen that the Q-routing algorithm can optimize the link load and significantly increase the link resource utilization, thereby improving the service request acceptance rate.

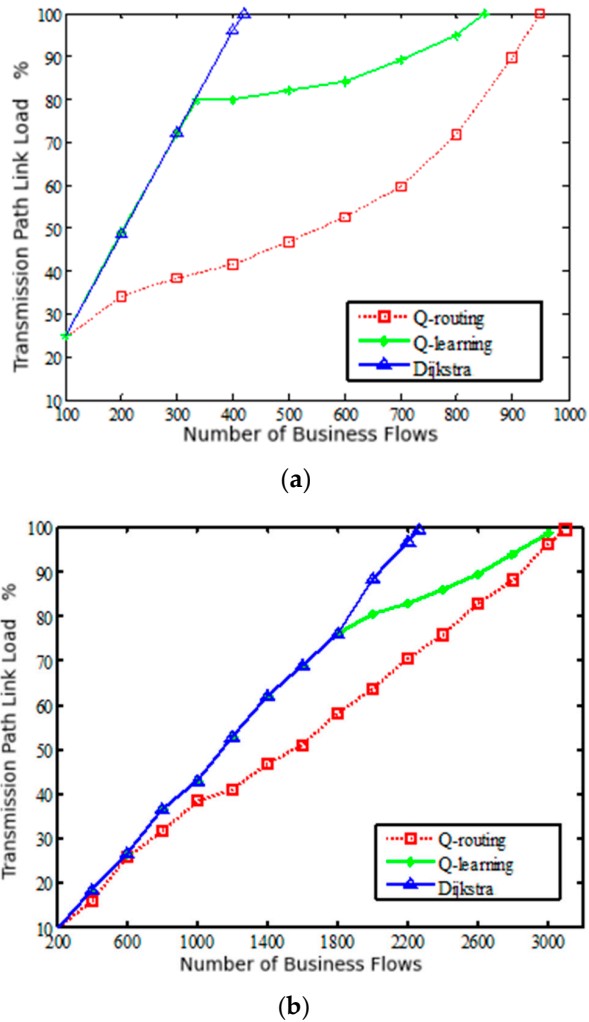

**Figure 14.** Comparison of service transmission path loads under different routing algorithms. (**a**) The source node and target node are fixed. (**b**) The source node and target node are random.

4.2.2. Average Transmission Delay of Service

Figure 15 describes the average transmission delay curves for delay-sensitive services under different algorithms. At this time, set the parameters in the Q-routing algorithm take $a_1$ to 0, $a_2$ to 1, and $a_3$ to 0. As shown in the figure, with the increase in service flow, the average transmission delay continues to rise, and the delay of the Q-routing routing algorithm is lower than that of the Dijkstra algorithm. Because for services with high delay requirements, when designing the reward function, the proportion of the reward for the delay is increased. Therefore, in the training phase, the transmission path with the lowest delay is tended to be selected. Before the link load reaches 80%, the transmission delay of the Q-learning algorithm and the Dijkstra algorithm are the same. Although it always forwards the data packet with the shortest number of hops, the link delay of each hop may not be the lowest, so the cumulative transmission delay is slightly higher than that of the Q-routing algorithm.

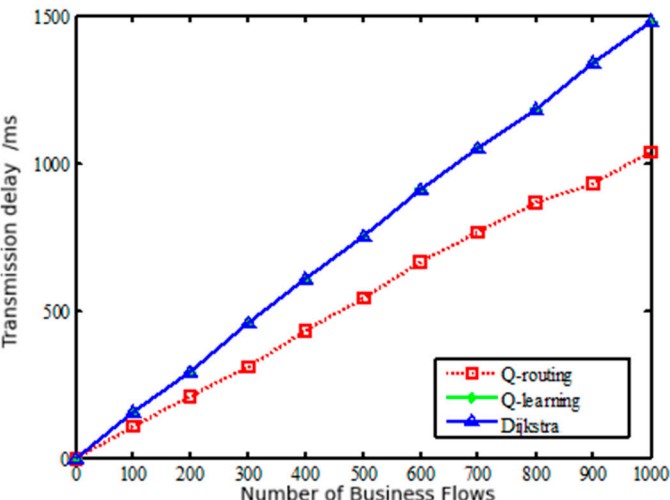

**Figure 15.** Comparison of transmission delays of delay-sensitive services under different routing algorithms.

Through the above comparison, we found that by setting the reward function in Q-routing, we can approach the optimal solution very well. In the process of comparing with other algorithms, the Q-routing routing algorithm can effectively balance the load and improve the request acceptance rate to a certain extent. It also has better performance than other algorithms for delay-sensitive services.

## 5. Conclusions

Routing technology is very important in inter-satellite link technology. Traditional algorithms can obtain only one shortest path, and is difficult to guarantee that a randomly discovered shortest path can meet the requirements of the service. The improved inter-satellite multipath algorithm combined with reinforcement learning, not only can obtain multiple shortest paths from the source node to all nodes in the wireless network, but also can adapt to the network dynamics in time and select the optimal path from all the shortest paths according to the load condition of the neighbor node when the global topology information of the network cannot be obtained. Simulation experiments show that the multipath routing algorithm for inter-satellite links based on load balancing can disperse network traffic pressure, avoid data traffic congestion during inter-satellite network system communication, have low latency and low packet loss rate, and have better network throughput performance. In addition, when the topology information cannot be obtained, the Q-routing algorithm with the link state considered, has obvious advantages in improving link utilization, and can better balance link bandwidth utilization and transmission delay compared to other algorithms.

**Author Contributions:** Writing—original draft preparation, Y.S.; writing—original draft, Z.Y.; supervision, X.Z., supervision, H.Z. All authors have read and agreed to the published version of the manuscript.

**Funding:** This research received no external funding.

**Institutional Review Board Statement:** Not applicable.

**Informed Consent Statement:** Informed consent was obtained from all subjects involved in the study.

**Data Availability Statement:** The data that support the findings are available on request from the corresponding authors. The data are not publicly available due to privacy.

**Acknowledgments:** This work was supported by Natural Science Foundation of China (92067101, 92067201), Program to Cultivate Middle-aged and Young Science Leaders of Universities of Jiangsu Province and Key R&D plan of Jiangsu Province (BE2021013-3).

**Conflicts of Interest:** The authors declare no conflict of interest.

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
