# Peer review of "An Adaptive Routing Algorithm for Inter-Satellite Networks Based on the Combination of Multipath Transmission and Q-Learning"

_processes, doi:10.3390/pr11010167_

Round 1
Reviewer 1 Report
The authors of the paper propose an algorithm for inter-satelite communication. First they apply multi-path routing algorithm and after reinforcement learning. The paper is very interesting and well written. It has practical application. All algorithm are described in details, including flow charts, and can be repeated by the reader. I have some small remarks concerning the references:
Reference [1] has not year of publication
list of references is too short.
Author Response
Thank you for your warm suggestions.
1.We have added the year of publication to Rerference[1].
2.We have cited more references related to our work.
Reviewer 2 Report
The paper is well written in term of technical overview. I just have some comments that might improve the overlook of the paper.
1. The author are suggested to define what is the mathematical operator such as ???????[?][?].
2. The authors are suggested to define (s)/w(s) < c(sn) /w(sn) in page 6 line 206.
3. The authors are suggested to follow the journal format for the tables, such as page 13.
4. The authors are suggested to consider the simulation scenario when distance between nodes is changing in time.
Thank you.
Author Response
Thank you for your warm suggestions.
1.We have revised the manuscript to define the ???????[?][?]
2.We have revised the manuscript to define the (s)/w(s) < c(sn) /w(sn)
3.We have changed the format of the table.
4.We think we have already taken the changing distance into consideration so that the link state is always changing.
Reviewer 3 Report
This paper considers an interesting problem that fits into the topics that deal with the Journal of Processes.
Author Response
Thank you for your kind comment.
Round 2
Reviewer 2 Report
many thanks for the authors for their sufficient response.